# Identification of Gene–Allele System Conferring Alkali-Tolerance at Seedling Stage in Northeast China Soybean Germplasm

**DOI:** 10.3390/ijms25052963

**Published:** 2024-03-04

**Authors:** Chunmei Zong, Jinming Zhao, Yanping Wang, Lei Wang, Zaoye Chen, Yuxin Qi, Yanfeng Bai, Wen Li, Wubin Wang, Haixiang Ren, Weiguang Du, Junyi Gai

**Affiliations:** 1Soybean Research Institute & MARA National Center for Soybean Improvement & MARA Key Laboratory of Biology and Genetic Improvement of Soybean (General) & State Key Laboratory for Crop Genetics and Germplasm Enhancement & State Innovation Platform for Integrated Production and Education in Soybean Bio-Breeding & Jiangsu Collaborative Innovation Center for Modern Crop Production, Nanjing Agricultural University, Nanjing 210095, China; zongcm@haas.cn (C.Z.); jmz3000@126.com (J.Z.); wl_as727@njau.edu.cn (L.W.); 2016201055@njau.edu.cn (Z.C.); wangbean@njau.edu.cn (W.W.); 2Mudanjiang Soybean Research and Development Center, Mudanjiang Branch of Heilongjiang Academy of Agricultural Sciences, Mudanjiang 157041, China; wyping1981@126.com (Y.W.); 6402296@163.com (Y.Q.); mdjnkybyf@163.com (Y.B.); liwenlevi@163.com (W.L.); rhx725@163.com (H.R.); weiguangdu@126.com (W.D.)

**Keywords:** alkali tolerance, gene–allele matrix, gene–allele sequence marker (GASM), gene function, Northeast China soybean germplasm population (NECSGP), optimal cross design, restricted two-stage multilocus genome-wide association study (RTM-GWAS), soybean (*Glycine max* (L.) Merr.)

## Abstract

Salinization of cultivated soils may result in either high salt levels or alkaline conditions, both of which stress crops and reduce performance. We sampled genotypes included in the Northeast China soybean germplasm population (NECSGP) to identify possible genes that affect tolerance to alkaline soil conditions. In this study, 361 soybean accessions collected in Northeast China were tested under 220 mM NaHCO_3_:Na_2_CO_3_ = 9:1 (pH = 9.8) to evaluate the alkali-tolerance (ATI) at the seedling stage in Mudanjiang, Heilongjiang, China. The restricted two-stage multi-locus model genome-wide association study (RTM-GWAS) with gene–allele sequences as markers (6503 GASMs) based on simplified genome resequencing (RAD-sequencing) was accomplished. From this analysis, 132 main effect candidate genes with 359 alleles and 35 Gene × Environment genes with 103 alleles were identified, explaining 90.93% and 2.80% of the seedling alkali-tolerance phenotypic variation, respectively. Genetic variability of ATI in NECSGP was observed primarily within subpopulations, especially in ecoregion B, from which 80% of ATI-tolerant accessions were screened out. The biological functions of 132 candidate genes were classified into eight functional categories (defense response, substance transport, regulation, metabolism-related, substance synthesis, biological process, plant development, and unknown function). From the ATI gene–allele system, six key genes–alleles were identified as starting points for further study on understanding the ATI gene network.

## 1. Introduction

Soybean [*Glycine max* (L.) Merr.] originated in ancient central China [1] and was then disseminated to Northeast China, including Heilongjiang, Jilin, Liaoning, and the eastern part of the Inner-Mongolia provinces, where cultivation and breeding flourished with its germplasm population formed and exchanged abroad [2]. At present, soybean cultivars currently grown in the Western Hemisphere and other parts of the world were derived mainly from the germplasm generated in Northeast China [3,4]. Therefore, the Northeast China soybean germplasm population (NECSGP) is an important gene pool not only in Northeast China but also in worldwide soybean production.

About one-third of the world’s irrigated land was affected by salinization [5,6]. The UNESCO and FAO indicated that the area of saline–alkali soil in the world exceeds 9.54 × 10^8^ hm^2^, of which about 60% is alkaline (high pH, dominated by sodium bicarbonate (NaHCO_3_) and sodium carbonate (Na_2_CO_3_)) [7]. Northeast China is one of the world’s three major soda saline–alkali soil areas [7], accounting for 39.59% of the total cultivated land (the proportion of saline–alkali soil area and cultivated land in Northeast China is 1.141 × 10^7^ hm^2^/2.882 × 10^7^ hm^2^ = 35.95%). Salinization caused two changes to the land, high salt content (more than 0.6%) and high alkaline or pH value (higher than 8.5). Alkali has a more serious effect on plants than neutral salt. Soil alkalization (including natural and anthropogenic) can reduce soil osmotic potential, cause plant ion imbalance, cause physiological process disorder, severely inhibit plant growth, and ultimately leads to a serious decline of seed yield and quality and can even lead to death [8].

In solving the plight of insufficient cultivated land, an effective way is to utilize saline–alkali tolerant cultivars [9]. Our previous research showed that there was no significant correlation between saline tolerance and alkali tolerance at the seedling stage (*r* = 0.044~0.071), indicating these two traits are different and relatively independent [10]. Compared to the research on plant saline tolerance, there are relatively few studies on plant alkali tolerance, especially for soybean. Wang et al. [11] indicated that in rice the gene regulatory network involved in alkali stress was more complex than that in saline stress. Alkali stress at the seedling stage may reduce the water absorption capacity of rice plants, and a high pH environment may affect plant homeostasis and metabolic balance, and ultimately affect plant growth and development [12]. Rogovska reported that high soil pH (pH > 8.2), accompanied by carbonate concentration stress from 2.5% to 30%, may cause iron deficiency chlorosis symptoms in soybeans and seed yield reduction up to 45% or more [13]. In searching for the genetic mechanism, four types of alkali-tolerance genes have been reported, i.e., intracellular pH regulation genes, organic-acid-metabolism-related genes, water-metabolism-related genes, and antioxidative-stress-related genes [14].

Soybean alkali tolerance is a complex quantitative trait. Linkage mapping based on biparental segregating populations is a common method in quantitative trait locus (QTL) mapping. With the advance of genome sequencing and the establishment of high-throughput single nucleotide polymorphism (SNP) technology [15], the genome-wide association study (GWAS) has played an important role in the genetic analysis of agronomic traits in soybean, including growth periods, 100-seed weight, seed oil content, seed protein content, sudden death syndrome, water use efficiency, etc. [16,17,18,19,20].

In order to make the GWAS better, the restricted two-stage multi-locus genome-wide association study (RTM-GWAS) was developed [21]. This study uses the genomic segment SNPLDB (SNP linkage disequilibrium block) with its multiple haplotypes as markers. Two stages were performed in the GWAS; in the first stage, general linear regression was conducted using the single locus model for pre-selection of markers, and in the second stage, stepwise regression using the multi-locus model was conducted with forward selection and backward deletion for identifying QTLs with their alleles. The total contribution of the selected QTLs is restricted by the experiment heritability value for controlling false positives and false negatives [21]. This procedure has been used to identify QTLs of soybean isoflavones, 100-seed weight, flowering date, etc. [22].

From the identified QTLs, the candidate genes located in/around the QTL were annotated. In this way, the genes to be selected for annotation are probably uncertain. To improve this, based on genome-wide sequencing, the gene–allele segments were used directly as genomic markers to replace the SNPLDBs. In this case, the identified QTL is in fact the gene marker itself, without additional requirement to infer from a QTL to a gene. This RTM-GWAS based on a gene–allele segment marker (GASM) is called GASM-RTM-GWAS, which has been used in identifying the gene–allele system for shade tolerance under soybean–maize inter-/relay-cropping in the southern China soybean germplasm [22]. These studies have demonstrated that the GASM-RTM-GWAS procedure was powerful and efficient in identifying a gene–allele system of complex traits and in predicting optimal crosses. Therefore, this method will be used in the present study on the gene–allele system of alkali-tolerance in the NECSGP.

The essence of crop breeding is the process of aggregating as many elite gene–alleles of interested trait(s) as possible into one variety, for which designing the optimal cross is the key to the success of breeding [23,24]. From the above, GASM-RTM-GWAS can provide all the genetic information on genes, their alleles, and the allele effects for each variety/accession in the germplasm/breeding population, from which all the crosses among the varieties/accessions can be predicted and practiced [25,26,27,28,29].

Excavating the alkali-tolerant resources and identifying elite genes/alleles in the Northeast China soybean germplasm is a key step to the development of alkali-tolerant cultivars for soybean production in Northeast China. This study aimed at the following targets: (i) on the basis of accurate evaluation of alkali-tolerance at the seedling stage of the NECSGP, using the alkali tolerance index (relative value of seedling dry weight under alkali stress to that under no alkali stress, ATI) as an indicator to excavate a germplasm with excellent seedling alkali tolerance; (ii) using GASM-RTM-GWAS to identify ATI genes with their alleles and allele effects conferring alkali-tolerance at the seedling stage of the NECSGP and to reveal the differentiation and genetic dynamics of the ATI gene–allele system among geographic subgroups; (iii) to predict the ATI recombination potential and to design the optimal combinations; and (iv) to identify the key ATI candidate genes and elite alleles for subsequent gene cloning and regulatory network research.

## 2. Results

### 2.1. Wide Variation of Alkali-Tolerance in the NECSGP

Through the expansion process of soybean in different subregions in Northeast China (from ERD to ERB, then to ERA and ERC, see the note of Table 1, Appendix A for details), a subregional germplasm with different ecological characteristics gradually formed. In the NECSGP, the comprehensive indicator, the ATI (the larger the ATI value the higher the tolerance to alkali conditions), showed a wide phenotypic variation under three environments, ranging from 0.36 to 0.86, with the genotypic coefficient of variation (GCV) equal to 19.96% and the heritability value as high as 98.43% (Table 1, Figure 1a). Analysis of variance showed that the variation among environments (E), genotypes (G), and *G* × *E* was significant (*p* < 0.01) (Appendix A), indicating that the ATI is mainly controlled by genetic effects but also affected by environment conditions. The ATI values at the seedling stage in each of the four ecological/geographic subregions (ERA−ERD) varied greatly and significantly, ranging from 0.46 to 0.86, 0.46 to 0.86, 0.37 to 0.83, and 0.39 to 0.85, with their genotypic coefficient of variation equal to 12.97%, 14.83%, 21.33%, and 16.95%, respectively, but without significant difference among the subpopulation averages. This suggests significant variation within each subregion but not among subregions (Table 1 and Appendix A).

From the NECSGP, the top ten alkali-tolerant and top ten alkali-sensitive varieties were selected. The former were represented by ATI values of 0.85 (0.83–0.86) whereas the latter had a ATI value of 0.42 (0.36–0.45) (Appendix A).

### 2.2. Gene–Allele System of Alkali-Tolerance in the NECSGP

In GASM-RTM-GWAS, the *G* × *E* model was adopted due to its significance based on an ANOVA. Using the 6503 GASMs in the RTM-GWAS procedure, in the first stage, 3246 GASMs were preselected for the second stage analysis, and then 132 GASMs/genes were detected for ATI (Table 2, Figure 1b,c). These 132 detected GASMs using the QEI model were distributed on all the 20 chromosomes, ranging from 2 GASMs on Chromosome 4 to 14 GASMs on Chromosome 18 (Figure 1b, Table 2). The genetic contribution (*R*^2^) of individual GASMs/genes ranged from 0.1% to 4.66% (Figure 1c); with *Glyma06g07980* having the largest contribution rate. Among these GASMs/genes, 28 were the large-contribution major genes (LC-major gene) with *R*^2^ values greater than 1.0% (Table 2 and Table 3). Of the 132 ATI genes, a total of 359 alleles were detected, with 2–7 alleles per gene (Figure 1e, Table 2), with *Glyma14g10780* having 7 alleles (the most). Among these alleles, 183 had positive effects ranging from 0.001 to 0.140, and 176 had negative effects ranging from −0.115 to −0.001 (Table 3). In total, 70 candidate genes were detected with more than two alleles (Table 2, Figure 1e).

Based on the GASM-RTM-GWAS results, the composition of the alkali-tolerance gene system in the NECSGP was explored (Table 2 and Table 3). The detected genes are composed of three types based on gene effects: genes with only genetic effects, genes with only *G* × *E* effects, and genes with both G and *G* × *E* effects (Table 2). For the ATI, 98.43% (heritability value) of the phenotypic variation (PV) was explained by main-effect genetic variation, of which 59.85% of the total *R*^2^ was accounted by 28 LC-major genes and 31.08% was accounted by 98 SC-major genes, with a total of 126 genes detected, with main-effect genes explaining 90.94% PV and the remaining 98.43 − 90.94 = 7.49% PV might be explained by the collective of unmapped minor genes. The *G* × *E* genes explained only 2.80% PV. The above results help us understand that alkali tolerance is a genetically complex trait.

### 2.3. ATI Gene–Allele Matrix as a Compact Genetic Structure of the NECSGP

All the detected ATI genes–alleles, with their effects from the 361 accessions, were organized into a gene–allele matrix (Figure 1h), which is a compact form of the genetic structure of the NECSGP as well as that of each accession. The matrix showed that all the accessions contained both positive and negative alleles (Figure 1d), indicating a great recombination potential for transgressive segregants in the population (Figure 1h).

In this study, the top 10 alkali-tolerant varieties and 10 alkali-sensitive varieties were selected for comparison to clarify the differences in their alkali tolerance gene–allele structures (Figure 1i, Appendix A). The number and effect values of positive alleles carried in alkali-tolerant accessions were more than those in alkali-sensitive accessions (69.4 and 1.83 vs. 64.1 and 1.62, respectively). Whereas the average number and absolute effect value of negative alleles carried in alkali-tolerant accessions is less than that of alkali-sensitive accessions (62.6 and −1.69 vs. 68.0 and −1.84, respectively). Thus, the alkali-tolerant varieties contain more elite positive alleles and less negative alleles than alkali-sensitive varieties. The major gene–allele constituents of the alkali-tolerant accessions were different from each other, whereas the sensitive varieties also contain elite alleles that can be utilized (Figure 1i, Appendix A).

### 2.4. Population Genetic Differentiation among the Four Subregions

The NECSGP gene–allele matrix was separated into its components of ERA–ERD. In Northeast China, ERD is the original ecoregion at the south, while ERB is the derived ecoregion based on ERD, and in turn, ERC and ERA were derived from ERB according to the dissemination paths of soybean. There were 200 alleles passed down from ERD to ERB, ERC, and ERA jointly, while 43 alleles in ERB, 27 alleles in ERA, and 0 alleles in ERC were different from those in ERD, indicating that the major changes were in ERB and ERA (Figure 2a).

In detailed changes, ERD contained 317 alleles (150 negative, 167 positive), which constituted basically the original genetic basis of the NECSGP (Table 4). Comparing ERB to ERD, in addition to 316 (153, 163) or 88.3% (315/358) of alleles being passed down, the allele number was enriched from 317 (150, 167) to 358 (176, 182) with 67 alleles changed, including 42 (22, 20) or 11.7% (42/358) of alleles that emerged (may including allele introduction) and only 1 (1, 0) or 0.3% (1/358) of alleles that were excluded. (Here, we could not distinguish if the new alleles were from allele emergence or allele introduction, so “emerged or introduced” was used, the same as below). Comparing ERC to ERD, in addition to 204 (100, 104) or 64.4% (204/317) of alleles being passed down, the allele number decreased to 210 (109, 101) with 119 alleles that changed, including 6 (1, 5) or 1.9% of alleles that emerged or were introduced and 113 (54, 59) or 35.6% (113/317) of alleles that were excluded. Comparing ERA to ERD, in addition to 289 (143, 146) or 91.2% (289/317) of alleles that were passed down, the allele number was similar with 316 (159, 157) with 55 (22, 33) alleles that changed, including 27 (11, 16) or 8.5% (27/317) of alleles that emerged or were introduced and 28 (11, 17) or 8.8% (28/317) of alleles that were excluded.

While comparing ERA to ERB, in addition to 315 (159, 156) or 88.0% (315/358) of alleles being passed down, the allele number decreased from 358 (176, 182) to 316 (159, 157), with 44 (20, 24) alleles having changed, including only 1 (0, 1) allele that emerged or was introduced and 43 (19, 24) or 12.0% (43/358) of alleles that were excluded. Comparing ERC to ERB, in addition to 210 (109, 101) or 58.7% (210/358) of alleles that were passed down, the allele number was decreased to 210 (101, 109) with 148 (74, 74) alleles that changed, including 0 or 0% of alleles that emerged or were introduced and 148 (74, 74) or 41.3% (148/358) of alleles that were excluded, a large number of alleles were not passed down to ERC.

There was no significant difference among subregion averages and ranges, but there was ATI gene–allele component changes which did not affect the sub-region alkali-tolerance average. The major change stage was from ERD to ERB. At this stage, the total ATI alleles increased, mainly new allele emergence happened, whereas allele exclusions were very few. Based on this, recombination also caused the progress. At the next stage, from ERB to ERA and ERC, the total ATE alleles decreased, especially for ERC, which was mainly due to the limited number of varieties introduced or developed. Almost no new allele emerged and both positive and negative alleles were excluded, indicating not very much effort has been put on alkali-tolerance breeding yet, especially in ERC which is the key subregion with an alkali problem. The above cases were supported by the fact that the selected best alkali-tolerant accessions were mainly from ERB (7 out of 8). Under this situation, the natural progress of alkali-tolerance from ERD to ERB was mainly due to allele emergence or introduction and emergence-based recombination. From ERD to ERA, 27 alleles increased, these likely have been inherited from ERB because very few alleles increased from ERB to ERA. In addition, both positive and negative alleles emerged and were excluded at almost all stages, despite the similar amount of alleles between the two sub-regions, this further demonstrates the selection direction for ATI tolerance throughout history was neutral or not directional.

### 2.5. Recombination Potential and Prediction of Optimal Crosses for Alkali-Tolerance Improvement in the NECSGP

Based on the ATI gene–allele matrix, a total of 64,980 biparental crosses among the 361 varieties were predicted in silico via the linkage model and independent assortment model, respectively, using the 75th percentile as the indicator with a population size of 2000 homozygous lines for each cross (Figure 1f). The prediction was also generated for each subpopulation (Table 5). Using the linkage model, the predicted 75th percentile of the NECSP was 0.75 on average, ranging from 0.47 to 1.03, with the best segregants over the highest variety exhibiting a 75th percentile value of 0.86. Those of the subregions were 0.97, 1.00, 0.92, and 0.93 for ERA-ERD, respectively, for their respective most tolerant variety. The predicted recombination potentials using the linkage model were similar to those generated using the independent assortment model, which suggests that there is no obvious linkage obstacle in the population, further confirming that breaking down the linkage for ATI in the NECSGP is not necessarily required.

From the prediction, the 10 top crosses are listed in Table 6, in which a total of eight parents were involved. Among the eight parents, Jiyu 93 (ATI = 0.81) was involved in five crosses and was the best, followed by Amsoy (four crosses) and Nenfeng 15 (three crosses). Among the ten crosses, the parental phenotypic values of ATI ranged from 0.81 to 0.86, while the predicted 75th percentile values of progenies ranged from 0.99 to 1.03, indicating that a great transgression might be expected from these crosses. These crosses are recommended for ATI breeding in Northeast China. In addition, seven out of the eight parents were from ERB, indicating more alkali-tolerance resources are held in this subpopulation.

### 2.6. The Candidate Gene System of Alkali-Tolerance Inferred from Identified GASMs

Based on the reference genome of Glyma.Wm82. a1. v1.1 (SoyBase, http://www.soybase.org, accessed on 17 May 2022), the gene ontology enrichment analysis indicated that the 132 ATI candidate genes were grouped into eight biological function categories. These include defense response (category I, 22 genes), material transports (category II, 4 genes), regulation related (category III, 11 genes), substance metabolism process (category IV, 25 genes), biosynthetic process (category V, 9 genes), biological process (category VI, 27 genes), development (category VII, 10 genes), and gene function unknown (category VIII, 14 genes), where category I, IV, and VI were the major categories (Appendix A, Figure 1g). Therefore, ATI is controlled by a great number of genes involving a series of biological functions, or a complex of gene systems.

In order to explore the interactive relationship among the identified ATI candidate genes, the protein–protein interaction (PPI) network was analyzed using the STRING procedure based on Arabidopsis homologous proteins (https://cn.string-db.org/, accessed on 25 May 2022) for the 132 ATI genes/proteins under medium confidence (interaction score 0.400). Since only 52 proteins were described in the STRING database, the present PPI results for the 52 proteins showed four major interaction groups (Figure 2b). The largest group contained 41 genes/proteins; the next was composed of 7 genes/proteins, while the two smallest groups each contained 2 genes/proteins. In each of the groups, the interacted genes/proteins were involved in different biological functions, indicating different functional genes were connected with each other. In Figure 2b, a gene may link with several different genes as a node/hub. There appeared to be a number of major hubs in the ATI gene system, which were involved in various biological functions. There were nine hub genes, including eight yellow hubs and one red hub in the largest group (Figure 2b(i)), i.e., *gATI.15.8* (category III, with 11 links), *gATI.6.3* (category III, with 6 links), *gATI.7.10* (category I, with 6 links), *gATI.13.4* (category VII, with 6 links), *gATI.2.15* (category IV, with 5 links), *gATI.7.7* (category V, with 5 links), *gATI.16.1* (category IV, with 5 links), *gATI.4.1* (category VI, with 4 links), *gATI.8.1* (category III, with 4 links), etc. Another hub gene was in the next largest group, i.e., *gATI.2.8* (category I, with 4 links) (Figure 2b(ii)). These hub genes were in the functional categories I, III, IV, V, VI, and VII, showing a broad interaction among the gene functional categories, especially three hubs with a total of 21 links involved in category III, indicating more genes linked with the regulation related function (Figure 2b). The PPI analysis demonstrated that the alkali-tolerance gene system was composed of interrelated gene networks. Under alkali stress, a great number of genes form complex regulatory networks, involving energy metabolism, antioxidants, ion transport, signal transduction, osmotic pressure balance, and others. However, the above ATI gene network information only involves part of the alkali-tolerant genes identified in this study, the complete alkali-tolerant gene network diagram needs to be further explored.

The key ATI genes were selected from the 132 identified ATI genes for future studies according to the following criteria: (i) genes with a large phenotypic contribution (*R*^2^ ≥ 1.00%); (ii) genes having an allele with a positive ATI effect; (iii) genes from the top ten accessions or top ten predicted crosses; and (iv) genes as a hub with multiple links to other genes. As a result, a total of 35 genes are summarized in Table 7, in which the genes that meet all four criteria, even if their allele effect is not high enough, were selected as a key ATI gene (bold and shaded in Table 7). Thus, the six key candidate genes are involved in four functional categories in response to alkali stress, including response and signal transmission genes (category I, *gATI.2.8* and *gATI.7.10*), positive regulation of transcription (category III, *gATI.6.3*), metabolism processes (category IV, *gATI.16.1* and *gATI.10.7*), and biological process (category VI, *gATI.4.1*). Among them, *gATI.6.3* and *gATI.7.10* are especially important because of their high phenotypic contribution (4.66% and 2.90%) and the alleles of *gATI.2.8.a3*, *gATI.6.3.a3*, and *gATI.16.1.a1* have the highest allele effects (0.079, 0.095, and 0.096). In addition to the six key genes, the other five hub genes with 4−11 links that existed in the top ten accessions and the top ten predicted best crosses were also listed in boldface in Table 7. They are *gATI.15.8* (11 links), *gATI.13.4* (6 links), *gATI.2.15* (5 links), *gATI.7.7* (5 links), and *gATI.8.1* (4 links). In fact, the nominated key genes are also hub genes; altogether, their broad interaction with other genes made the gene an interrelated network. All the nominated six key genes, even the 6 + 5 = 11 hub genes with their alleles, can be used as the starting points to reveal the interrelationship among ATI genes–alleles as well as the major gene-sources for ATI improvement.

## 3. Discussion

### 3.1. Advances in Exploring the Genetic System of Alkali-Tolerance

Regarding the study of the genetic system of alkali-tolerance, early researchers did not distinguish alkali tolerance from salt tolerance; they called them salt–alkali tolerance or simply salt tolerance. In this way, Abel [30] identified a tolerance gene *NCL* and allele *ncl*. Shao et al. [31] used underground dilute brine (mainly composed of Na^+^, Cl^−^, HCO^3−^, CO_3_^2−^, etc.) to analyze the tolerance performance in tolerant × sensitive crosses and concluded that the tolerance was controlled by a single gene. In fact, this was not salt tolerance, but tolerance to a mixture of salt and alkali conditions. As indicated in the present Introduction Section, salt tolerance and alkali tolerance are quite different tolerance traits.

Subsequent researchers tried to analyze the genetic system of alkali tolerance using marker technology. Tuyen [6] used linkage mapping with 180 mM NaHCO_3_ as an alkali stress induction to identify and locate an alkali tolerance QTL close to *Satt447* on chromosome 13 in two recombinant inbred line (RIL) populations and then used residual heterozygous lines (RHLs) to narrow down the QTL between Satt447 and GM17-11.6, in which four candidate genes were annotated [32]. Zhang [33] used composite interval mapping (CIM) with 180 mM NaHCO_3_ to map QTLs of seven alkali-tolerance indicators in two RIL populations, with the results of 22 and 19 QTLs detected, respectively. Obviously, these results involve only a few QTLs per trait even though a quantitative inheritance procedure was used.

For exploring the alkali tolerance in a germplasm population, Zhang et al. [34] used a genome-wide association study procedure for 257 soybean varieties under 10 mM Na_2_CO_3_ stress, in which 16–18 QTLs in a total of 86 QTLs for five indicators were identified. This result indicates that alkali-tolerance is a quantitatively inherited trait rather than an oligogenic trait. But the procedure used was not powerful enough because the identified QTL number and their phenotypic contribution was less than it should be. In contrast, the present study identified 132 genes with 359 allele effects and a total phenotypic contribution as high as 90.94% for only a single trait. This indicates that alkali tolerance is a genetically complex trait, while GASM-RTM-GWAS helped to achieve this understanding.

### 3.2. The Main Genetic Motivator of Alkali-Tolerance in the NECSGP by Using the GASM-RTM-GWAS Procedure

In the NECSGP, the major genetic changes in ATI were from ERD to ERB; the major ATI-tolerance resources (eight of top ten accessions) were from ERB; and the major genetic changes that happened in ERB were due to new alleles that emerged or were introduced (22 negative + 20 positive alleles) rather than alleles that were excluded (only 1 negative + 0 positive alleles). While the increased alleles (11 negative + 16 positive alleles) in ERA were inherited from ERB, rather than being newly emerged/introduced. This implies that, for ATI genetic changes from ERD to ERB in the NECSGP, the major genetic motivator was allele emergence or introduction, followed by recombination among alleles based on newly increased plus inherited from ERD.

Among the 42 emerged or introduced alleles in ERB, 20 positive ones should be important to ATI improvement, rather than the 22 negative ones. Appendix A showed the 20 emerged/introduced new alleles on 20 corresponding loci. Their locus contribution ranged from 0.002 to 2.00, the allele effect ranged from 0.004 to 0.085, 13 new alleles existed in the top ten accessions and best crosses, and all the alleles were involved in all the functional categories except category I. Among the 20 new alleles, 3 were important alleles, i.e., *gATI.2.8.a4* (*R*^2^ 1.28, effect 0.085), *gATI.4.1.a2* (*R*^2^ 2.00, effect 0.042), and *gATI.7.7.a2* (*R*^2^ 0.06, effect 0.074). These implies that in the evolutionary process, the newly emerged/introduced alleles happened in almost all the functional processes, and in the NECSGP, the newly emerged/introduced alleles have accounted for a great contribution to alkali-tolerance improvement in ERB and ERA.

Direct comparison from the gene–allele matrix revealed that allele inheritance, emergence/introduction, elimination, and recombination are the four basic driving forces of ATI evolution. This provides an endless source of breeding progress. The direct comparison procedure in studying the motivator of population evolution was suggested originally by Fu et al. [35], which is based on the relative thorough identification of the genes/QTLs and their corresponding alleles in an evolutionary or geographic population. This procedure has been used effectively by Liu et al. [2] and Su et al. [22].

In the present study, it was achieved by using the GASM-RTM-GWAS procedure [21], with GASMs as markers such that the 132 ATI genes with their 359 alleles were identified directly. The GASM-RTM-GWAS procedure features two major innovations [22]. The first is using multiple allele markers to fit the requirement of multiple alleles in the germplasm population or breeding population (previously SNPLDB and at present GASM), so it can detect gene loci as well as their alleles. The second is to minimize the false positives and false negatives through the following ways: (i) the multilocus model to avoid neighboring loci interference; (ii) limit of the total phenotypic contributions to less than the heritability value; (iii) two-stage analysis to reduce noise from excessive markers; (iv) incorporating the top 10 eigenvectors and their eigenvalues of the marker–haplotype (gene–allele) matrix into the linear model as covariates for population structure correction; (v) precise experiment design to reduce experiment error on a plot-based analysis to keep a precise significance test; and (vi) a reasonable significance level for the multi-locus model test that is equivalent to the Bonferroni criterion (0.05/m) in the single locus model [21].

As indicated in the present study, 132 genes with their 359 alleles were identified; GASM-RTM-GWAS is more powerful and efficient than the other procedures, such as SNP-MLM-GWAS and SNPLDB-RTM-GWAS [22]. Thus, the GASM-RTM-GWAS procedure fits especially well with the germplasm populations, with multiple alleles per locus, enabling an understanding of the complete gene–allele system in population evolutionary study.

### 3.3. Breeding Strategy for Alkali-Tolerance Improvement in the NECSGP

From GASM-RTM-GWAS, the obtained ATI gene–allele matrix provides a way to predict the optimal crosses. In the NECSGP, out of the total 64,980 possible predicted crosses, 5696 (8.77%) could provide offspring lines with an ATI value greater than 0.86; the best accession in the population with the top ten crosses might be close to 1 or may not be affected by alkali stress. This indicates a promising potential in the population. This is in fact the result of genome-wide potential based on whole genome gene–allele identification. It may be understood as genome-wide cross design or another kind of genomic selection based on Peleman and van der Voort [25], which is different from Meuwissen et al.’s [36] genomic selection based on the estimated breeding value for a set of traits from a training population. As we understand, the former is more applicable to plant breeding because the breeding procedure usually includes two steps of selection, i.e., selection for optimal crosses and selection for optimal progenies, with the former step being more important. This strategy is based on the elite allele of each QTL/gene from the whole genome analysis of germplasm resources [28] and has been successfully applied in the optimization design of multiple traits such as days to flowering [35]. As shown in Table 6, the standard deviation among progenies of each combination was around 20%, indicating that the segregation among the progenies in a cross was extensive and potential.

In addition, it seems that in the present population, due to the newly emerged/introduced alleles in ERB, the recombination potential of the NECSGP was much improved. That implies more ATI tolerant materials/gene resources need to be introduced or created/induced for further ATI improvement. To design new optimal crosses based on the increased germplasm, additional resequencing for the additional germplasm will be needed. To save work, a gene chip can be developed and used based on the previous gene–allele sequence information. This requires the extension of the genetic dissection to the increased germplasm according to the original whole genome gene–allele information.

## 4. Materials and Methods

### 4.1. Plant Materials

A total of 361 accessions, including 243, 80, 20, 13, and 5 accessions collected from Heilongjiang, Jilin, Liaoning, and Inner Mongolia, respectively, with their maturity group in MG 000–MG III, were used in the present study (Appendix A). According to Fu et al. [35], these accessions were representative of the four ecoregions in Northeast China, i.e., 61, 230, 8, and 62 from ERA, ERB, ERC, and ERD (Appendix A), respectively, where the original soybean ecoregion was ERD, then disseminated to ERB and then to ERA and ERC (Table 1, Appendix A).

### 4.2. Experimental Design and Evaluation of ATI at the Seedling Stage

In the spring and summer of 2018 and spring of 2020, the 361 accessions were tested in a greenhouse at the Mudanjiang Branch of the Heilongjiang Academy of Agricultural Sciences (Mudanjiang, China; 129°52″ E, 44°43″ N) with the ATIs evaluated in paper rolls in cups (Appendix A). A randomized block design with 3 replications was used for each season. In each season, each accession was set such that each treatment and control paralleled each other as a unit/plot with guarding rows around the whole experiment. For each unit/plot, 6 well-matured uniform seeds were put in the middle of a filter paper (30 × 20 cm), then rolled, 1 roll in one plastic cup (5.3 × 7.8 × 11.2 cm) filled with 360 mL 1/2 Hoagland’s nutrient solution, two cups for one unit/plot, one for treatment, and another for the check. At the first leaf (V1) stage, 3 healthy plants with uniform growth were selected for further testing in each cup. The stress treatment started at the V2 stage, with the alkali stress treatment using a mixed solution based on 1/2 Hoagland’s nutrient solution added with NaHCO_3_:Na_2_CO_3_ (9:1) into a concentration of 220 mM with pH = 9.8, while another control cup used 1/2 Hoagland’s nutrient solution only. The fresh solution was changed for both the stress treatment and the control every 2 days (on the 3rd, 5th, and 7th day) at about noon time. Then, on the 8th day, the whole plant, including roots, stems, and leaves (some had wilted already), was harvested and put in a paper bag. The harvests were killed with conditions of 109 °C for 10 min, dried under 80 °C to a constant weight, and then weighed. The ATI was calculated as the ratio of dry treatment weight divided by the dry control weight. Under alkali stress, all varieties show different degrees of reduced plant height, shortened internodes, weak and slow growth, and reduced dry weight. However, as the alkali stress continued to a later stage, the cells lose water, the roots become dry and hard, the leaves turn yellow, the stem withered, and finally the whole plant died.

### 4.3. Method of Statistical Analysis

Descriptive statistics, variance analysis, correlation analysis, and chi-square test of phenotypic data were performed using SAS 9.4 (SAS Institute Inc., Cary, NC, USA). The statistical model of multienvironment phenotype data is
$$y_{ijk}=\mu +\alpha_{i}+\beta_{j}+{\left(\alpha \beta \right)}_{ij}+\gamma_{k(j)}+\varepsilon_{ijk},$$
where μ is the population mean, $\alpha_{i}$ is the effect of *i*-th variety, $\beta_{j}$ is the effect of the *j*-th environment, ${\left(\alpha \beta \right)}_{ij}$ is the *i*-th variety by the *j*-th environment interaction effect, $\gamma_{k(j)}$ is the effect of the *k*-th replication within the *j*-th environment, and $\varepsilon_{ijk}$ is random error. The heritability (*h*^2^) and genetic variation coefficient (*GCV*) are calculated from $h^{2}={\hat{\sigma }}_{g}^{2}/{(\hat{\sigma }}_{g}^{2}+{\hat{\sigma }}_{ge}^{2}/n{+\hat{\sigma }}_{\varepsilon }^{2}/rn),\;GCV(\%)={\hat{\sigma }}_{g}/\mu \times 100\%$, where ${\hat{\sigma }}_{g}^{2}$, ${\hat{\sigma }}_{ge}^{2}$, and ${\hat{\sigma }}_{\varepsilon }^{2}$ are the genotype, genotype–environment interaction, and error variance, respectively; *n* is environment number, *r* is experimental repetition number, and *μ* is the population mean [37].

### 4.4. Genotyping and Assembly of GASM Markers

RAD sequencing of 361 accessions was conducted at BGI Tech, Shenzhen, China [38]. The genomic DNA from young leaves was extracted according to the conventional CTAB method [39]. Paired-end sequencing was carried out on the HisSeq 2000 platform with the multiplex shotgun genotyping method [40]. A total of 1227.23 million paired-end reads with a length of 100 bp (including a 6 bp index) were generated, and the mean sequencing depth and coverage rate was 4.21 fold and 3.42%, respectively. All sequence reads were mapped to a reference genome (Wm82.v.1.1) [41] with SOAP2 [42]. SNP calling was performed using RealSFS software [43] (http://128.32.118.212/thorfinn/realSFS/, accessed on 12 March 2019) and filtered by a minimum missing and heterozygosity of ≤20% and a minor allele frequency (MAF) of ≥0.01. SNPs were imputed using fastPHASE software (1.3.0) [44].

All SNPs in a gene were directly assembled into a gene–allele sequence-maker (GASM) using RTM-GWAS [21], with the highest frequency alleles substitution for alleles variation of 0.01. Gene assembly was performed according to the gene annotation file (SoyBase_Glycine.max_Glyma1.1_gene.txt). When there was an overlap at the gene position, the gene with a longer length was retained. The assembled genes were used to construct GASMs in the NECSGP; a total of 6503 GASMs, containing 14,847 alleles, were identified, the GASMs are distributed on 20 chromosomes (Appendix A).

### 4.5. Identification of the ATI Gene–Allele System Using GASM-RTM-GWAS

Based on the GASMs, the RTM-GWAS procedure was used to identify ATI genes and their corresponding alleles. In the first stage, a single-locus model association analysis was conducted to preselect GASMs at *p* ≤ 0.05. Of the 6503 GASMs, 3246 were preselected. In the second stage, stepwise regression with a *G* × *E* model featuring forward selection and backward elimination using the multilocus model was applied to the preselected 3246 GASM markers to identify genes with their alleles at a model test significance level *p* ≤ 0.05 which is equivalent to a Bonferroni criterion of *p* ≤ 0.05/marker number (3246 here). At both stages, the top 10 eigenvectors and their eigenvalues were incorporated into the linear model as covariates for population structure correction [21]. Finally, the dataset of the estimated allele effects for all the main effect GASMs/genes was organized into a gene–allele matrix, which was separated into 4 ecoregion submatrices. In addition, a matrix was established for the top ten tolerant and top ten sensitive accessions. Based on the matrices, the allele changes, including being inherited, emerged, or introduced, and excluded alleles were counted for comparisons among the subregions to evaluate the importance of genetic motivators in the geographic dissemination.

### 4.6. Recombination Potential Prediction and Optimal Cross Design for ATI in the NECSGP

Based on the ATI gene–allele matrix, all possible 64,980 single crosses among the 361 accessions were generated in silico using both linkage and independent assortment models using RTM-GWAS [21]. For each cross, the predicted genotypic ATI value was calculated based on 2000 continuously inbred progenies derived from F_2_ individuals. The 75th percentile of a cross was used as its predicted cross value. The best 10 crosses were treated as the ATI optimal crosses of the population.

### 4.7. Functional Annotation and Interaction of Candidate Genes

Based on the reference genome Wm82.a1.v1 proposed by Schmutz et al. [41], the identified alkali-tolerant gene system was functionally annotated according to SoyBase (http://www.soybase.org, accessed on 18 May 2022).

In order to explore the interactions of each gene in the alkaline-tolerance gene system, the protein sequences of these genes were copied from the Phytozome database (http://phytozome.jgi.doe.gov, accessed on 24 May 2022) and then submitted to the “*Glycine max*” section of STRING (https://cn.string-db.org/, accessed on 25 May 2022), visualizing PPI using STRING analysis based on Arabidopsis homologous proteins.

## 5. Conclusions

The alkali-tolerance index (ATI) at the seedling stage of the NECSGP, composed of 361 accessions, varied widely from 0.36 to 0.86 with an average of 0.65, but was not significantly different among averages of the four ecoregions, from 0.64 in ERC to 0.69 in ERC. In total, 132 ATI genes with their 359 alleles were identified using the innovative GASM-RTM-GWAS procedure. The major tolerant accessions were found in ERB, due to both new allele emergence/introduction and old allele exclusion as well as recombination among the present alleles. The ATI recombination potential was predicted up to 1.0 (without stress influence) in the population, in which the 10 best crosses with transgression were recommended for ATI breeding programs. Based on the above results, six key genes with their best alleles were selected for further functional study.

## Figures and Tables

**Figure 1 ijms-25-02963-f001:**
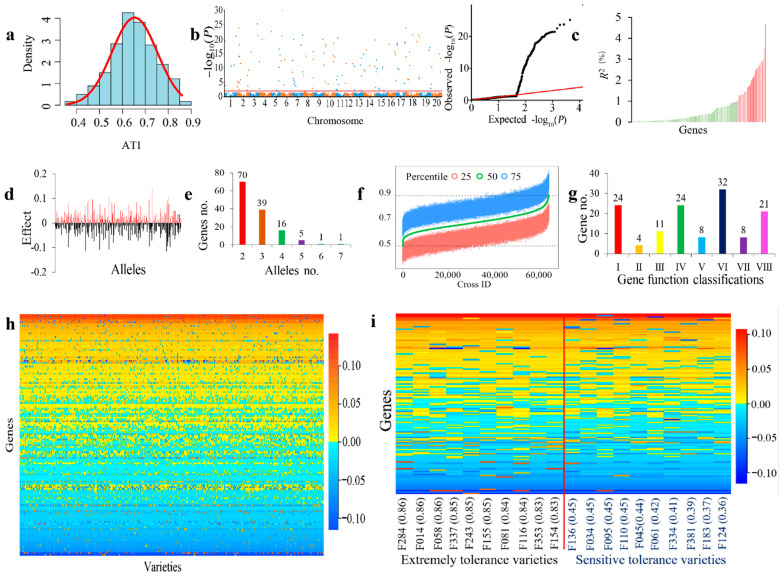
ATI gene–allele system identified in the NECSGP using GASM-RTM-GWAS. (**a**): Frequency distribution of the ATI at the seedling stage averaged over three environments in the NECSGP. (**b**): Manhattan (left) and Q-Q plot (right) of ATI candidate genes identified in the NECSGP. The horizontal solid red line indicates the threshold of *p* < 0.05, the upper colored dots are the significant candidate genes mapped in different chromosomes. (**c**) Phenotypic contribution of genes detected for ATI. The vertical axis indicates genetic contribution of a gene, while for the horizontal axis, the genes with a green color have an *R*^2^ value ≥ 1.0% and for those with a red color *R*^2^ < 1.0%. (**d**) Distribution of allele effect values of ATL alleles of each gene. (**e**) Frequency distribution of allele number per ATI gene. (**f**) Recombination potential prediction of ATI in the NECSGP based on the linkage model. (**g**) Gene ontology classifications of ATI in the NECSGP; I: Defense response; II: Material transport; III: Regulation related; IV: Metabolic process; V: Biosynthetic process; VI: Biological process; VII: Development; and VIII: unknown. (**h**) ATI gene–allele matrix for the NECSGP. The horizontal axis represents the varieties of each subregion which are arranged in ascending order based on their phenotypic value, while the vertical axis represents the alleles on the chromosomes. The allelic effects are represented by different colors; warm colors represent positive values and cool colors represent negative values. The darker the color, the greater the absolute value of the ATI value. (**i**) ATI gene–allele matrix of the top 10 tolerant and top ten sensitive varieties at the seedling stage.

**Figure 2 ijms-25-02963-f002:**
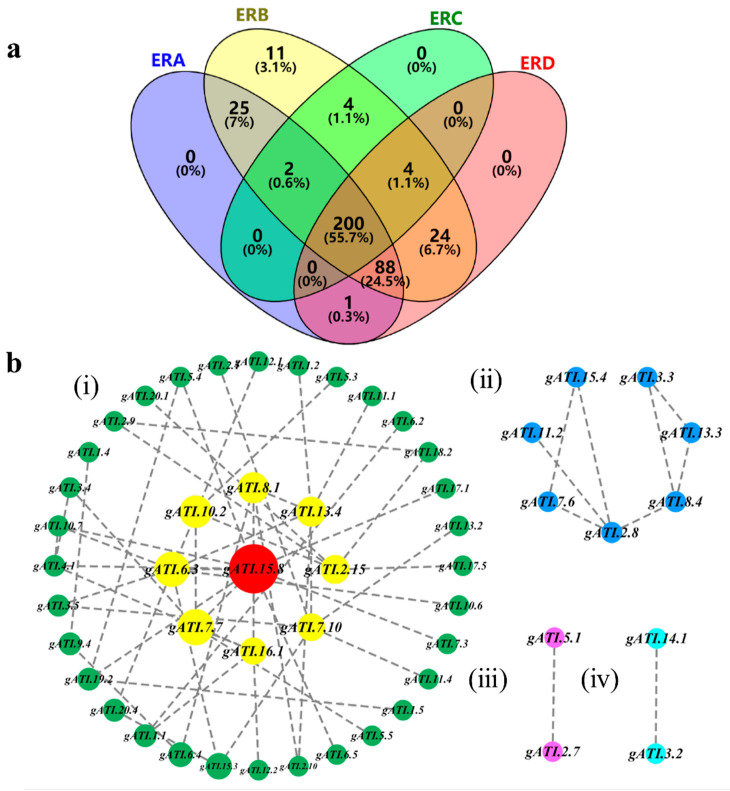
Venn diagram of allele number shared by subregions and protein–protein interaction network of the ATI gene system in the NECSGP. Note: (**a**) Venn diagram of allele number shared by subregions. (**b**) Protein–protein interaction (PPI) networks. (**i**) The largest PPI network, containing 41 genes/proteins; the red and yellow nodes in the middle part with more links are hub genes. (**ii**) the blue nodes comprise the second largest network with 7 genes/proteins. (**iii**,**iv**) the deep pink nodes and bright green nodes, represent two small networks, each with only two genes/proteins.

**Table 1 ijms-25-02963-t001:** Frequency distribution of ATI in the NECSGP.

ENV. ^†^	Mid-Point Classic											SUM	Mean	Range	*GCV* ^‡^ (%)	*h*^2^ ^§^ (%)
	0.38	0.43	0.48	0.53	0.58	0.63	0.68	0.73	0.78	0.83	0.88					
Total	3	9	16	27	48	50	85	48	48	19	7	361	0.65	0.36~0.86	19.96	98.43
ERA			3	3	9	10	16	7	10	2	1	61	0.64 a *	0.46~0.86	12.97	94.62
ERB	1	6	6	12	30	24	58	30	33	16	14	230	0.66 a	0.46~0.86	14.83	96.57
ERC	1						2	1	1	1	2	8	0.69 a	0.37~0.83	21.33	93.71
ERD	1	3	2	4	11	5	14	7	8	3	4	62	0.64	0.39~0.85	16.95	95.36

Note: ^†^ Env.: ERA: Ecoregion of Heilongjiang valleys; ERB: Ecoregion of Songhuajiang valleys; ERC: Ecoregion of Nenjiang valleys; ERD: Ecoregion of Liaohe valleys (Please See Appendix A for details). The dissemination path of soybeans in Northeast China was from ERD to ERB, then from ERB to ERA and ERC. The upper right corner shows the location of the Northeast region in China. ^‡^
*GCV*: Genetic Coefficient of Variation. ^§^
*h*^2^: heritability value calculated from the ANOVA. * a: Significance level at 0.05 level compared to ERD.

**Table 2 ijms-25-02963-t002:** The ATI gene–allele system identified via GASM-RTM-GWAS in the NECSGP.

Gene Code	Gene Name	AN	*R*^2^ (%)		Gene Code	Gene Name	AN	*R*^2^ (%)	
			Main Effect	*G* × *E*				Main Effect	*G* × *E*
*gATI.1.1*	*Glyma01g02580*	2	1.32		*gATI.10.8*	*Glyma10g37420*	2	0.56	0.04
*gATI.1.2*	*Glyma01g04515*	3	0.09		*gATI.11.1*	*Glyma11g01253*	3	0.05	
*gATI.1.3*	*Glyma01g22861*	2	0.19		*gATI.11.2*	*Glyma11g03580*	2	0.11	
*gATI.1.4*	*Glyma01g30320*	2	0.28		*gATI.11.3*	*Glyma11g07830*	4	0.71	
*gATI.1.5*	*Glyma01g36070*	2	1.69		*gATI.11.4*	*Glyma11g15140*	2	0.01	0.05
*gATI.2.1*	*Glyma02g00280*	3	0.05	0.04	*gATI.11.5*	*Glyma11g27510*	2	0.1	
*gATI.2.2*	*Glyma02g05640*	3	0.08		*gATI.12.1*	*Glyma12g03180*	3	0.10	
*gATI.2.3*	*Glyma02g06730*	5	3.48	0.07	*gATI.12.2*	*Glyma12g08010*	2	0.26	
*gATI.2.4*	*Glyma02g08790*	2	0.03		*gATI.12.3*	*Glyma12g30080*	3	0.24	
*gATI.2.5*	*Glyma02g09130*	3	0.70		*gATI.13.1*	*Glyma13g00490*	3	0.38	0.04
*gATI.2.6*	*Glyma02g09240*	3	0.41		*gATI.13.2*	*Glyma13g01900*	2	0.04	
*gATI.2.7*	*Glyma02g11151*	4	0.78	0.08	*gATI.13.3*	*Glyma13g23910*	3	0.64	
*gATI.2.8*	*Glyma02g11335*	4	1.28		*gATI.13.4*	*Glyma13g29225*	3	0.12	
*gATI.2.9*	*Glyma02g14175*	2	0.12		*gATI.13.5*	*Glyma13g29360*	2	0.45	0.02
*gATI.2.10*	*Glyma02g15470*	3	0.28		*gATI.13.6*	*Glyma13g29520*	2	0.04	
*gATI.2.11*	*Glyma02g16850*	3	0.39		*gATI.13.7*	*Glyma13g40690*	2	1.03	
*gATI.2.12*	*Glyma02g37010*	2	0.05		*gATI.13.8*	*Glyma13g42650*	3	0.24	0.04
*gATI.2.13*	*Glyma02g38673*	3	0.19		*gATI.14.1*	*Glyma14g04260*	3	0.14	
*gATI.2.14*	*Glyma02g40220*	3	0.06		*gATI.14.2*	*Glyma14g10780*	7	0.98	
*gATI.2.15*	*Glyma02g44350*	2	0.24		*gATI.14.3*	*Glyma14g20110*	2	0.31	
*gATI.3.1*	*Glyma03g07890*	2	0.74		*gATI.14.4*	*Glyma14g36130*	2	0.25	
*gATI.3.2*	*Glyma03g27770*	2	2.23		*gATI.14.5*	*Glyma14g37280*	3	2.84	0.05
*gATI.3.3*	*Glyma03g30270*	4	2.14	0.04	*gATI.14.6*	*Glyma14g38720*	2	0.05	
*gATI.3.4*	*Glyma03g36720*	2		0.04	*gATI.15.1*	*Glyma15g02310*	2	0.35	
*gATI.3.5*	*Glyma03g37221*	2	0.33	0.08	*gATI.15.2*	*Glyma15g04006*	3	0.32	
*gATI.4.1*	*Glyma04g10720*	2	2.00		*gATI.15.3*	*Glyma15g07590*	3	0.33	
*gATI.4.2*	*Glyma04g43300*	2	2.88		*gATI.15.4*	*Glyma15g16830*	2	0.03	
*gATI.5.1*	*Glyma05g19630*	2	0.02	0.02	*gATI.15.5*	*Glyma15g19900*	2		0.03
*gATI.5.2*	*Glyma05g24760*	3	0.09		*gATI.15.6*	*Glyma15g27480*	5	0.36	
*gATI.5.3*	*Glyma05g27300*	2	0.94		*gATI.15.7*	*Glyma15g27750*	2	0.37	
*gATI.5.4*	*Glyma05g27690*	5	0.20	0.08	*gATI.15.8*	*Glyma15g32540*	2		0.03
*gATI.5.5*	*Glyma05g32890*	4	0.81		*gATI.16.1*	*Glyma16g08960*	2	1.43	
*gATI.6.1*	*Glyma06g01490*	2	0.49	0.03	*gATI.16.2*	*Glyma16g28270*	3	0.91	
*gATI.6.2*	*Glyma06g05300*	3	1.82		*gATI.16.3*	*Glyma16g32650*	5	0.89	
*gATI.6.3*	*Glyma06g07980*	3	4.66		*gATI.16.4*	*Glyma16g33831*	4	0.10	
*gATI.6.4*	*Glyma06g17410*	2	0.08		*gATI.17.1*	*Glyma17g07120*	3	1.66	
*gATI.6.5*	*Glyma06g19756*	2	0.20		*gATI.17.2*	*Glyma17g08230*	2	0.95	
*gATI.6.6*	*Glyma06g47010*	4	0.14		*gATI.17.3*	*Glyma17g10650*	4	0.39	
*gATI.7.1*	*Glyma07g00400*	4	0.73		*gATI.17.4*	*Glyma17g15720*	2	0.03	
*gATI.7.2*	*Glyma07g05620*	4	2.45	0.03	*gATI.17.5*	*Glyma17g16831*	3	0.11	0.11
*gATI.7.3*	*Glyma07g06640*	2	0.10		*gATI.17.6*	*Glyma17g33020*	2	0.09	
*gATI.7.4*	*Glyma07g07360*	6	0.12	0.05	*gATI.17.7*	*Glyma17g33930*	2	2.56	
*gATI.7.5*	*Glyma07g08290*	2		0.05	*gATI.17.8*	*Glyma17g36130*	2	1.46	0.08
*gATI.7.6*	*Glyma07g10280*	2	0.08		*gATI.18.1*	*Glyma18g01490*	2	2.71	
*gATI.7.7*	*Glyma07g14234*	2	0.06		*gATI.18.2*	*Glyma18g03930*	2		0.02
*gATI.7.8*	*Glyma07g31130*	3	1.48	0.12	*gATI.18.3*	*Glyma18g03975*	2	0.68	
*gATI.7.9*	*Glyma07g37810*	2	0.16		*gATI.18.4*	*Glyma18g04870*	2	0.35	
*gATI.7.10*	*Glyma07g38180*	3	2.90		*gATI.18.5*	*Glyma18g11512*	4	0.15	
*gATI.8.1*	*Glyma08g04620*	4	0.18		*gATI.18.6*	*Glyma18g15001*	3	0.03	0.09
*gATI.8.2*	*Glyma08g10001*	3	0.02		*gATI.18.7*	*Glyma18g16761*	3	0.27	
*gATI.8.3*	*Glyma08g18320*	2	0.38		*gATI.18.8*	*Glyma18g16780*	2	0.71	
*gATI.8.4*	*Glyma08g42810*	4	0.05		*gATI.18.9*	*Glyma18g28130*	2	3.00	
*gATI.8.5*	*Glyma08g45301*	3	0.65		*gATI.18.10*	*Glyma18g40780*	3	1.71	
*gATI.8.6*	*Glyma08g45501*	2	0.13		*gATI.18.11*	*Glyma18g46101*	3	2.62	
*gATI.8.7*	*Glyma08g45610*	2	0.72	0.15	*gATI.18.12*	*Glyma18g49450*	2	0.12	
*gATI.9.1*	*Glyma09g12180*	3	0.03		*gATI.18.13*	*Glyma18g50670*	4	0.17	0.04
*gATI.9.2*	*Glyma09g15860*	2	0.03		*gATI.18.14*	*Glyma18g53285*	3	2.06	0.07
*gATI.9.3*	*Glyma09g33220*	4	2.52	0.05	*gATI.19.1*	*Glyma19g31900*	4	1.28	
*gATI.9.4*	*Glyma09g40690*	3	0.68		*gATI.19.2*	*Glyma19g35820*	2	0.78	
*gATI.9.5*	*Glyma09g41821*	3	0.38	0.03	*gATI.19.3*	*Glyma19g42710*	2		0.04
*gATI.10.1*	*Glyma10g06480*	2	0.08		*gATI.19.4*	*Glyma19g43880*	2	0.06	
*gATI.10.2*	*Glyma10g06600*	2	0.80		*gATI.20.1*	*Glyma20g01460*	2	0.22	0.01
*gATI.10.3*	*Glyma10g07601*	2	1.32		*gATI.20.2*	*Glyma20g24740*	5	0.95	0.06
*gATI.10.4*	*Glyma10g14620*	2	0.06		*gATI.20.3*	*Glyma20g30360*	2	0.97	
*gATI.10.5*	*Glyma10g30100*	3	0.75	0.04	*gATI.20.4*	*Glyma20g30870*	2	0.17	
*gATI.10.6*	*Glyma10g30320*	2	0.27						
*gATI.10.7*	*Glyma10g31560*	2	1.32	0.02	Total	132	^a^ 359 (2.72)	90.94	2.80

Note: Gene code, such as *gATI.1.1*, where ATI means alkali tolerance, the first .1 represents Chromosome 1 and the second .1 represents its order on the chromosome according to its physical position. The position corresponds to the Williams 82 reference genome version 1 (Wm82.a1). ATI: alkali tolerance index. *R*^2^: contribution to phenotypic variance of a gene. Main effect: main effect gene; *G* × *E*: gene–environment interaction. ^a^: The number before the parentheses represents the number of total alleles, and the number in the parentheses represents the average number of alleles per gene.

**Table 3 ijms-25-02963-t003:** Summary of ATI gene–allele information in the NECSGP.

**Gene**	**Main Effect %**	***G* × *E* %**
Significant	90.94 (126, 0.01~4.66)	2.80 (35, 0.01~0.15)
LC major Gene	59.86 (28, 1.03~4.66)	
SC major Gene	31.08 (98, 0.01~0.98)	2.80 (35, 0.01~0.15)
Unmapped minor Gene	7.49	
Total genetic contribution *h*^2^	98.43	2.80
**Allele**	**Main Effect**	***G* × *E***
Positive allele	131 (0.0004~0.1420)	52 (0.001~0.107)
Negative allele	125 (−0.1150~−0.0010)	51 (−0.108~−0.001)
Total	256 (−0.1150~0.1420)	103 (−0.108~0.107)

Note: In the upper part for genes, LC-major Gene: large-contribution major Gene with genetic contribution (*R*^2^) greater than 1.0%. SC-major Gene, small-contribution major Gene with *R*^2^ less than 1.0%. In columns of Main effect and *G* × *E*, the number outside parentheses is the total *R*^2^ of the corresponding genes, the first number in the parentheses is the number of genes, the second is a range of *R*^2^ for the individual genes. The *R*^2^ of the unmapped minor gene is calculated from the total contribution (*h*^2^)—the contribution of main effect genes. In the lower part for alleles, the number outside of the parentheses is the number of alleles while the number in the parentheses is the range of the allele effect.

**Table 4 ijms-25-02963-t004:** The allele changes in subregions compared to ERD and then to ERB.

Contrast	Total Alleles	Alleles Changed			
		Inherited	Emerged/Introduced	Excluded	Changed
ERB vs. ERD	358 (176, 182) vs. 317 (150, 167)	316 (153, 163)	42 (22, 20)	1 (1, 0)	43 (23, 20)
ERC vs. ERD	210 (109, 101) vs. 317 (150, 167)	204 (100, 104)	6 (1, 5)	113 (54, 59)	119 (55, 64)
ERA vs. ERD	316 (159, 157) vs. 317 (150, 167)	289 (145, 144)	27 (11, 16)	28 (14, 14)	55 (25, 30)
RA vs. ERB	316 (159, 157) vs. 358 (176, 182)	315 (156, 159)	1 (1, 0)	43 (19, 24)	44 (20, 24)
ERC vs. ERB	210 (109, 101) vs. 358 (176, 182)	210 (101, 109)	0	148 (74, 74)	148 (74, 74)

Note: In the total alleles column, the number outside the parentheses is the number of total alleles and the number in parentheses is the number of negative and positive alleles, respectively. Emerged/introduced means alleles that emerged or were introduced compared to alleles in ERD or ERB; the number in parentheses is the number of negative and positive alleles, respectively, while the number outside of parentheses is the sum of them. In the changed column is the sum of emerged/introduce alleles and excluded alleles.

**Table 5 ijms-25-02963-t005:** Predicted recombination potential of ATI in the NECSGP and its subregions.

Region/ Subregion	ATI Range of Accessions	No. of Predicted Crosses	Predicted via the Linkage Model				Predicted via the Independent Assortment Model			
			Mean	Range	High Tolerance Cross No. (ATI > 0.86)	Percentage (%)	Mean	Range	High Tolerance Cross No. (ATI > 0.86)	Percentage (%)
Total	0.36~0.86	64,980	0.75	0.47~1.03	4388	6.75	0.75	0.48~0.98	4580	7.05
ERA	0.46~0.86	2775	0.74	0.53~0.97	150	5.41	0.74	0.54~0.96	145	5.23
ERB	0.46~0.86	32,896	0.75	0.47~1.00	2096	6.37	0.76	0.48~0.98	2190	6.66
ERC	0.37~0.83	28	0.79	0.62~0.92	6	21.42	0.79	0.62~0.94	6	21.43
ERD	0.39~0.85	210	0.76	0.55~0.93	19	9.05	0.76	0.56~0.93	26	12.38

Note: The predicted value is the 75th percentile. Percentage (%): the percentage of high-tolerance crosses to the total number of crosses.

**Table 6 ijms-25-02963-t006:** The predicted top ten crosses for ATI expressed in 75th percentile values of progenies (Based on linkage model).

P_1_				P_2_				Predicted Cross (75th Percentile)		
Code	Name	MG	ATI	Code	Name	MG	ATI	Mean of Parents	*SD* (%)	Predicted ATI
F134	Nenfeng 15	I	0.82	F81	Amsoy	I	0.84	0.84	26.07	1.03
F243	Kenjian 35	0	0.85	F81	Amsoy	I	0.84	0.85	21.48	1.00
F243	Kenjian 35	0	0.85	F81	Amsoy	I	0.84	0.85	21.48	1.00
F104	Jiyu 93	I	0.81	F284	Mufeng No. 3	0	0.86	0.83	23.12	0.99
F104	Jiyu 93	I	0.81	F134	Nenfeng 15	I	0.82	0.82	25.49	0.99
F104	Jiyu 93	I	0.81	F387	Jiyu 86	III	0.83	0.82	23.11	0.99
F134	Nenfeng 15	I	0.82	F284	Mufeng No. 3	0	0.86	0.84	20.9	0.99
F104	Jiyu 93	I	0.81	F243	Kenjian 35	0	0.85	0.82	23.53	0.99
F233	Kennong 24	I	0.82	F81	Amsoy	I	0.84	0.84	20.06	0.99
F104	Jiyu 93	I	0.81	F135	Nenfeng 17	0	0.81	0.82	24.08	0.99

Note: P_1_ and P_2_ represent the parents of the predicted optimal cross; MG is the maturity group to which the parents belong; SD represents the standard deviation of the cross with 10,000 progenies per cross.

**Table 7 ijms-25-02963-t007:** The nominated key ATI candidate genes with their alleles at the seedling stage in the NECSGP.

Gene Code	Gene Name	*R* ^2^	Superior Allele	Allele Effect	In Accessions and Parents	No. of Links	Gene Function and Classification
*gATI.1.1*	*Glyma01g02580*	1.32	*gATI.1.1.a2*	0.016	√	2	
*gATI.1.5*	*Glyma01g36070*	1.69	*gATI.1.5.a1*	0.024	√	1	
*gATI.2.3*	*Glyma02g06730*	3.48	*gATI.2.3.a2*	0.027	√		
* **gATI.2.8** *	* **Glyma02g11335** *	1.28	*gATI.2.8.a3* *	0.079	√	4	Cell–cell signaling (I)
* **gATI.2.15** *	* **Glyma02g44350** *	0.24	*gATI.2.15.a1*	0.043	√	5	Acetyl-CoA metabolic process (IV)
*gATI.3.2*	*Glyma03g27770*	2.23	*gATI.3.2.a1*	0.023	√	1	
*gATI.3.3*	*Glyma03g30270*	2.14	*gATI.3.3.a1*	0.051	√	2	
* **gATI.4.1** *	* **Glyma04g10720** *	2.00	*gATI.4.1.a1* *	0.021	√	4	Biological process (VI)
*gATI.4.2*	*Glyma04g43300*	2.88	*gATI.4.2.a2*	0.048	√		
*gATI.6.2*	*Glyma06g05300*	1.82	*gATI.6.2.a3*	0.083	√	1	
* **gATI.6.3** *	* **Glyma06g07980** *	4.66	*gATI.6.3.a3*	0.095	√	6	Positive regulation of transcription (III)
*gATI.7.2*	*Glyma07g05620*	2.45	*gATI.7.2.a4*	0.053	√		
* **gATI.7.7** *	* **Glyma07g14234** *	0.06	*gATI.7.7.a2* *	0.042	√	5	Metabolic process (IV)
*gATI.7.8*	*Glyma07g31130*	1.48	*gATI.7.8.a2*	0.077	√		
* **gATI.7.10** *	* **Glyma07g38180** *	2.90	*ATI.7.10.a3*	0.028	√	6	Response to wounding (I)
* **gATI.8.1** *	* **Glyma08g04620** *	0.18	*gATI.8.1.a2*	0.039	√	4	Protein dephosphorylation (III)
*gATI.8.4*	*Glyma08g42810*	0.05	*gATI.8.4.a1*	0.017	√	3	
*gATI.9.3*	*Glyma09g33220*	2.52	*gATI.9.3.a3*	0.082	√		
*gATI.10.2*	*Glyma10g06600*	0.80	*gATI.10.2.a2*	0.023	√	3	
*gATI.10.3*	*Glyma10g07601*	1.32	*gATI.10.3.a1*	0.015	√		
* **gATI.10.7** *	* **Glyma10g31560** *	1.32	*gATI.10.7.a1*	0.018	√	3	Ubiquitin-dependent protein catabolic process (IV)
* **gATI.13.4** *	* **Glyma13g29225** *	0.12	*gATI.13.4.a1*	0.029	√	6	NA (VII)
*gATI.13.7*	*Glyma13g40690*	1.03	*gATI.13.7.a1*	0.012	√		
*gATI.14.5*	*Glyma14g37280*	2.84	*gATI.14.5.a3*	0.068	NA		
*gATI.15.8*	*Glyma15g32540*	(0.03)	*gATI.15.8.a1*	0.005	√	11	Translation (III)
* **gATI.16.1** *	* **Glyma16g08960** *	1.43	*gATI.16.1.a1*	0.096	√	5	Fatty acid beta-oxidation (IV)
*gATI.17.1*	*Glyma17g07120*	1.66	*gATI.17.1.a1*	0.033	√	1	
*gATI.17.7*	*Glyma17g33930*	2.56	*gATI.17.7.a2*	0.015	√		
*gATI.17.8*	*Glyma17g36130*	1.46	*gATI.17.8.a1*	0.061	√		
*gATI.18.1*	*Glyma18g01490*	2.71	*gATI.18.1.a1*	0.045	√		
*gATI.18.9*	*Glyma18g28130*	3.00	*gATI.18.9.a2*	0.076	√		
*gATI.18.10*	*Glyma18g40780*	1.71	*gATI.18.10.a1*	0.108	√		
*gATI.18.11*	*Glyma18g46101*	2.62	*gATI.18.11.a1*	0.078	√		
*gATI.18.14*	*Glyma18g53285*	2.06	*gATI.18.14.a3*	0.048	NA		
*gATI.19.1*	*Glyma19g31900*	1.28	*gATI.19.1.a2*	0.05	√		

Note: Gene code: candidate genes with *R*^2^ ≥ 1.00% selected from 132 alkali tolerance candidate genes. Superior alleles: alleles with large effect values (Please See Appendix A for details). In accessions and parents: genes existed in the best accessions and best predicted optimal crosses. No. of links: the number of links of a node in the PPI network. The six genes in bold and shaded are the nominated key candidate genes, and the genes in bold are hub genes. Gene function and classification: I: defense response; II: material transport; III: regulation of related; IV: metabolic process; V: biosynthetic process; VI: biological process; and VII: unknown function. * the three alleles were newly emerged or introduced in ERB.

## Data Availability

The data presented in this study are available in the Appendix A.

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
