# Peer review of "Identification of Gene–Allele System Conferring Alkali-Tolerance at Seedling Stage in Northeast China Soybean Germplasm"

_ijms, 2024, doi:10.3390/ijms25052963_

Round 1
Reviewer 1 Report
Comments and Suggestions for Authors
This investigation for genes that mitigate tolerance to high alkaline soils in soybean was well conceived, planned, and executed. The results appear to be a trove of important information for soybean breeders and stress tolerance investigators. The presentation of the manuscript, particularly the language, must be improved (see examples provided below for the Abstract) before the it is recommended for publication.
Abstract
Lines 17-18) “After salinization, the soil may challenge the crops with saline-stress or alkali stress, which are different stresses with the latter characterized with a high pH (>8.5) value.” Suggest change to “Salinization of cultivated soils may result in either high salt levels or alkaline conditions, both of which stress crops and reduce performance.”
Line 17) “Alkali-stress is one of the major abiotic stresses.” Delete.
Lines 18-21) “Identifying the gene-allele system of alkali tolerance at seedling stage of the Northeast China Soybean Germplasm Population (NECSGP) is the prerequisite for soybean production in Northeast China.” Suggest change to “We screened genotypes included in the Northeast China Soybean Germplasm Population (NECSGP) to identify possible genes that affect tolerance to alkaline soil conditions.”
Line 25) “…was carried out, from which 132 main…” change to “…was accomplished. From this analysis 132 main…”.
Lines 27-31) “In NECSGP, the ATI genetic changes happened mainly from ecoregion D to B rather than to C and A, during which the major genetic motivators were new allele emergence/introduction combined with further recombination. . In NECSGP, eight of the top ten alkali-tolerant accessions were from ecoregion B, in which, the major elite alleles and the predicted optimal crosses located.” Suggest change to “Genetic variability within NECSGP for ATI was observed primarily within subpopulations for ecoregions D to B and 80% of ATI tolerant accessions were from ecoregion B.”
Line 32) “…eight categories…” change to “…eight phenotypic categories…”.
Keywords
The string of terms should be alphabetized.
Body of manuscript
Refer to suggestions made for the Abstract in revising the language to improve the clarity of presentation.
Comments on the Quality of English LanguageSee Comments for Editors
Author Response
General comment: This investigation for genes that mitigate tolerance to high alkaline soils in soybean was well conceived, planned, and executed. The results appear to be a trove of important information for soybean breeders and stress tolerance investigators. The presentation of the manuscript, particularly the language, must be improved (see examples provided below for the Abstract) before the it is recommended for publication:
Answer: Thanks for the general comment. We have consulted our faculties in plant genetics/breeding and plant molecular biology to help for revising the manuscript.
Comment 1: Lines 17-18, “After salinization, the soil may challenge the crops with saline-stress or alkali stress, which are different stresses with the latter characterized with a high pH (>8.5) value.” Suggest change to “Salinization of cultivated soils may result in either high salt levels or alkaline conditions, both of which stress crops and reduce performance.”.
Answer: Thanks for the suggestion. The suggested expression has been adopted.
Comment 2: Line 17, “Alkali-stress is one of the major abiotic stresses.” Delete
Answer: According to the comment, this sentence has been deleted.
Comment 3: Lines 18-21, “Identifying the gene-allele system of alkali tolerance at seedling stage of the Northeast China Soybean Germplasm Population (NECSGP) is the prerequisite for soybean production in Northeast China.” Suggest change to “We screened genotypes included in the Northeast China Soybean Germplasm Population (NECSGP) to identify possible genes that affect tolerance to alkaline soil conditions.”.
Answer: According to the comment, we have changed it into “We sampled genotypes included in the Northeast China Soybean Germplasm Population (NECSGP) to identify possible genes that affect tolerance to alkaline soil conditions”. Here we use “sampled” instead of “screened” because the material used in this study had not been screened yet.
Comment 4: Line 25) “…was carried out, from which 132 main…” change to “…was accomplished. From this analysis 132 main…”.
Answer: Thanks for the comment. We have replaced it with the reviewer's expression.
Comment 5: Lines 27-31) “In NECSGP, the ATI genetic changes happened mainly from ecoregion D to B rather than to C and A, during which the major genetic motivators were new allele emergence/introduction combined with further recombination. In NECSGP, eight of the top ten alkali-tolerant accessions were from ecoregion B, in which, the major elite alleles and the predicted optimal crosses located.” Suggest change to “Genetic variability within NECSGP for ATI was observed primarily within subpopulations for ecoregions D to B and 80% of ATI tolerant accessions were from ecoregion B.”
Answer: Thanks for the suggestion, we have replaced it with the reviewer’s expression with modification “Genetic variability of ATI in NECSGP was observed primarily within subpopulations, especially in ecoregion B, from where 80% of ATI tolerant accessions were screened out”.
Comment 6: Line 32) “…eight categories…” change to “…eight phenotypic categories…”.
Answer: Thanks for the suggestion. It has been change to “…eight functional categories…” because the classification is on gene functions.
Comment 7: Keywords. The string of terms should be alphabetized.
Answer: According to the comment, the keywords have been sorted alphabetically.
Reviewer 2 Report
Comments and Suggestions for Authors
Identification of a system of gene alleles for alkali resistance in the germplasm population is necessary to increase the resistance and productivity of plants, soybeans in particular. The authors have done a great job. Methods correspond to goals. The work is well written from a scientific point of view, the English needs correction. During the work, some comments arose. The file with comments is attached.
Sincerely.

English needs correction
Author Response
General comment: Identification of a system of gene alleles for alkali resistance in the germplasm population is necessary to increase the resistance and productivity of plants, soybeans in particular. The authors have done a great job. Methods correspond to goals. The work is well written from a scientific point of view, the English needs correction. During the work, some comments arose. The file with comments is attached.
Answer: Thanks for the general comment. We have invited an experienced English-speaking colleague to help us to improve the writings of the entire manuscript.
Comment 1: Lines 17-18: Commented [A1]: the phrase is misleading.
Answer: Thanks for the comment. We have adopted Reviewer#1’s suggested expression with some modifications.
Comment 2: Lines 27-31: Commented [A2]: To understand all ecoregions you need to read the work; I think here you should not focus on the transition, but rather talk about alleles. the phrase is unclear
Answer: As indicated in Reviewer#1’ comment 5, these two sentences have been deleted and replaced with another sentence. Please refer to that answer. In followed sentences, the ATI gene functions and key alleles were mentioned.
Comment 3: Lines 48: Commented [A3]: Sub-regions can be mentioned for understanding
Answer: Thanks for the comment. We have supplemented the spread of soybean in various ecological sub-regions in Northeast China to facilitate readers' understanding.
Comment 4: Lines 56: Commented [A4]: indicate sources of salinity, both natural and anthropogenic.
Answer: We added the causes of soil alkalization, “Soil alkalization (including natural and anthropogenic)”.
Comment 5: Lines 63: Commented [A5]: this article (the 10th reference) could not be found to read.
Answer: The 10th reference is the doctoral dissertation of the author (Chunmei Zong).
Comment 6: Lines 137-138: Commented [A6]: What could this mean? (That means significant variation within each sub-region but not among them)
Answer: Yes, it means that the alkali tolerance of germplasm varies greatly within each sub-region, but the difference among sub-region averages is not significant, please see Table 1, Table S2 for details.
Comment 7: Lines 460-478: Commented [A7]: The section 3.1 of discussion,it is worth talking about the correlation of genes and resistance and yield in more detail.
Answer: Thanks for the comment to remind us to link alkali-tolerance with yield and other traits. Previous studies have discovered a few genes related to alkali-tolerance in other crops but not in soybeans. Unfortunately, we did not have enough data yet on the relationship between alkali-tolerance and other traits, such as yield and resistances. We will try to do so and would leave it in the future. Currently, in section 3.1 of discussion, the discussion is mainly on the genomic selection procedure based on the GWAS results for transgressive segregation of alkali-tolerance.
Round 2
Reviewer 1 Report
Comments and Suggestions for Authors
This paper summarizes the outcomes of a large study undertaken to ascertain alkali tolerance in soybean germplasm from northeastern China aimed at estimating the numbers of genes and alleles and to infer the geophysical progression of germplasm between subregions. The study was ambitious and yield very compelling results that will be of great interest and utility to plant geneticists and breeders. Because the English in the manuscript was not acceptable and the reviewer was incompetent to evaluate some of the statistical procedures incorporated in the paper, the reviewer did not fully evaluate the information presented in the Discussion and Materials and Methods sections. Specific comments and suggestions are as follows:
Abstract
Line 13) “…in zones diverging for one or the other parameter.” Change to “…under different environments.”.
Line 15) “…platform, and…” change to “…platform and…”.
Line 16) “…additionally…” suggest change to “…further…”.
Line 17) “…analysis, performed…” change to “…analysis performed…”.
Line 19) “…unraveled…” suggest change to “…identified…”.
Lines 20-21) “Spatial variability maps of the expression level of key berry ripening genes in showed consistent patterns, aligned with the vineyard vigor map.” This sentence must be rewritten to be meaningful.
Lines 21-23) “These insights suggest that berries from different vigor zones present distinct molecular maturation programs hence showing potential in predicting spatial variability in fruit quality.” Reviewer believes the authors are trying to conclude “These results demonstrate a relationship between the expression of specific genes and grape berry maturation status and suggest that transcriptome may be a valuable tool for management of vineyard yield and quality.”.
Keywords
Words or phrases used in the list should be alphabetized.
Introduction
Lines 39-41) “The cultivated soybean [Glycine max (L.) Merr.] was originated in ancient China [1] which disseminated to Northeast China and became a major crop there with its germplasm population formed…” this sentence is awkwardly worded and the reviewer suggests the following revision: “Soybean (Glycine max L. Merr.) was domesticated in ancient China [1] then disseminated to modern day Mongolia and Manchuria where cultivation and breeding flourished…”.
Lines 41-44) “Through the expansion process of soybean in different sub-regions in Northeast China (from ERD to ERB then to ERA and ERC, see the note of Table 1, supplementary Figure 2 for details), sub-regional germplasm with different ecological characteristics gradually formed.” This sentence should be deleted and/or moved to the Results or Discussion sections.
Lines 44-46) “It was commonly known that the major germplasms of soybeans in North and South Americas were traced directly or indirectly from those of Northeast China soybeans [3, 4].” Change to “Soybean cultivars currently grown in the Western Hemisphere and other parts of the world were derived from the germplasm generated in Northeastern China [3,4].”
Lines 46-48) The authors invoke an existing breeding population (The Northeast China Soybean Germplasm Population) as a valuable resource to all soybean breeders. This would be an ideal opportunity to discuss the origins and access to the resource.
From this point to the end of the manuscript the reviewer addressed only issues of content, not language or grammar.
Line 54) “…accounting for 39.59% of the total cultivated land…” the authors should explain how this figure was imputed.
Line 91) “This procedure has…” the reviewer suggests the start of a new paragraph here. The foregoing discussion of methodology is interesting but should be abridged or moved to the Discussion section.
Overall the Introduction contains too much technical information that should be abridged or moved to other sections such as Materials and Methods, Results, or Discussion.
Results
Line 128) “…extremely significant…” the universally accepted term is “…highly significant…”.
Line 132) “…0.46~0.86, 0.46~0.86, 0.37~0.83, and 0.39~0.85…” to what do these ranges correspond to? The text indicates that these ranges were measured from different germplasm “subregion” sources but these are not identified so are meaningless.
Line 179) “…both effects…” reviewer presumes this refers to G and GxE effects? This should be stated in the text.
Line 179-185) “For ATI, 98.43% (heritability value) of the phenotypic variation (PV) was explained by main effect genetic variation, in which 59.85% for the total R2 of 28 LC-major genes, 31.08% for 98 SC-major genes, in a total of 126 detected main effect genes explaining 90.94% PV, and the remained 98.43-90.94=7.49% PV might be explained by the collective unmapped minor genes. While the G×E genes explained only 2.80 % PV, which was very small.” The above results make us understand that alkali tolerance is a genetically complex trait.” The reviewer is not a quantitative geneticist so was not able to verify the validity of these estimates. If the authors have not already done so, these estimates should be confirmed by such an expert.
Lines 194-195) “The same is for below.” Table and figure legends should be complete and sufficient for each item described. Legends should not reference other legends.
Table 3) The presentation of data is confusing in this table. For the “main effects” and “allele effects” the number of alleles is depicted in different ways. The authors should reformat the table to be clearer.
Line 210) “…breakthrough…” a more commonly used term by plant breeders would be “…transgressive…”.
Lines 226-227) “…according to the dissemination paths of soybean.” The references concerning the physical movement of Glycine germplasm in northeastern China from the Introduction should be reiterated here to document this statement and serve as the foundation for the following paragraphs.
Lines 250-251) “…but there was ATI genetic structure changes…” The reviewer is confused about what the structural changes were; the authors should elaborate.
Lines 260-262) “The previous progress in alkali-tolerance was mainly a by-path without a scientific breeding plan.” Delete or move to the Discussion section.
Line 272) Legend for Figure 2; “…number in different…” Reviewer believes the authors intend to say “…number of alleles specific to subregion…”?
Table 4) This table is full of figures, parentheses, varying units, etc. and is too confusing for the reviewer to follow. The authors should try to simplify the table to portray the basic message and move most of the details to an addendum.
Line 328) “…(inadequate information in alkali-tolerance)…” Delete or explain.
Line 349) “…the key ones were nominated…” nominated how? By whom? Reviewer is confused by “nominated”.
Lines 372-373) “…candidate genes with R2≥1.00% selected from 132 alkali tolerance candidate genes…” R2 of 1% seems quite small and arbitrary. Is there a precedent in the literature concerning the threshold for minimum correlation coefficient?
Discussion
Line 437) “In the present study,…” New paragraph.
Line 451) “As indicated in the present study,…” New paragraph.
Comments on the Quality of English Language
The English in this manuscript must be dramatically improved for this manuscript to be publishable.
Author Response
RESPONSE TO THE COMMENTS FROM REVIEWER #2
General comment: This paper summarizes the outcomes of a large study undertaken to ascertain alkali tolerance in soybean germplasm from northeastern China aimed at estimating the numbers of genes and alleles and to infer the geophysical progression of germplasm between sub-regions. The study was ambitious and yield very compelling results that will be of great interest and utility to plant geneticists and breeders. Because the English in the manuscript was not acceptable and the reviewer was incompetent to evaluate some of the statistical procedures incorporated in the paper, the reviewer did not fully evaluate the information presented in the Discussion and Materials and Methods sections. Specific comments and suggestions are as follows.
Answer: Thanks for the general comment. We have invited an English-speaking colleague to help us to improve the writings of the manuscript. As for the statistical procedures, especially RTM-GWAS, used in the present manuscript, we cited a number of citation published in diverse journal to support our procedures and methods. However, some of those were deleted for keeping not very many self-citations according to the journal’s rules.
Abstract
Comment 1: Line 13) “…in zones diverging for one or the other parameter.” change to “…under different environments.”.
Comment 2: Line 15) “…platform, and…” change to “…platform and…”.
Comment 3: Line 16) “…additionally…” suggest change to “…further…”.
Comment 4: Line 17) “…analysis, performed…” change to “…analysis performed…”.
Comment 5: Line 19) “…unraveled…” suggest change to “…identified…”.
Comment 6: Lines 20-21) “Spatial variability maps of the expression level of key berry ripening genes in showed consistent patterns, aligned with the vineyard vigor map.” This sentence must be rewritten to be meaningful.
Comment 7: Lines 21-23) “These insights suggest that berries from different vigor zones present distinct molecular maturation programs hence showing potential in predicting spatial variability in fruit quality.” Reviewer believes the authors are trying to conclude “These results demonstrate a relationship between the expression of specific genes and grape berry maturation status and suggest that transcriptome may be a valuable tool for management of vineyard yield and quality”.
Answer: It seems that comment 1 to comment 7 are not comments for the present manuscript because these comments involving grape berries.
Keywords
Comment 8: Keywords: Words or phrases used in the list should be alphabetized.
Answer: According to the comment, the keywords have been sorted alphabetically.
Introduction
Comment 9: Introduction: Lines 39-41, “The cultivated soybean [Glycine max (L.) Merr.] was originated in ancient China [1] which disseminated to Northeast China and became a major crop there with its germplasm population formed…” this sentence is awkwardly worded and the reviewer suggests the following revision: “Soybean (Glycine max L. Merr.) was domesticated in ancient China [1] then disseminated to modern day Mongolia and Manchuria where cultivation and breeding flourished…”.
Answer: According to the comment, this sentence has been changed as “Soybean (Glycine max L. Merr.) was domesticated in ancient central China [1] then disseminated to Northeast China, including Heilongjiang, Jilin, Liaoning and eastern part of Inner-Mongolia provinces, where cultivation and breeding flourished with its soybean germplasm population formed and exchanged abroad”. Mongolia is another country while Manchuria was the previous name of Northeast China used by the Japanese invader, it is not suitable to be used here.
Comment 10: Lines 41-44, “Through the expansion process of soybean in different sub-regions in Northeast China (from ERD to ERB then to ERA and ERC, see the note of Table 1, supplementary Figure 2 for details), sub-regional germplasm with different ecological characteristics gradually formed.” This sentence should be deleted and/or moved to the Results or Discussion sections.
Answer: According to the comment, this sentence has been moved to Results section at the beginning of 2.1 sub-section.
Comment 11: Lines 44-46, “It was commonly known that the major germplasms of soybeans in North and South Americas were traced directly or indirectly from those of Northeast China soybeans [3, 4].” Change to “Soybean cultivars currently grown in the Western Hemisphere and other parts of the world were derived from the germplasm generated in Northeastern China [3,4].”
Answer: According to the comment, this sentence has been changed to “At present, soybean cultivars currently grown in the Western Hemisphere and other parts of the world were derived mainly from the germplasm generated in Northeast China [3,4].”
Comment 12: Lines 46-48, The authors invoke an existing breeding population (The Northeast China Soybean Germplasm Population) as a valuable resource to all soybean breeders. This would be an ideal opportunity to discuss the origins and access to the resource.
Answer: According to the suggestion, at the beginning of the first paragraph, we mentioned “Soybean [Glycine max (L.) Merr.] was originated in ancient central China [1] then disseminated to Northeast China”, to indicate its origin from central China. Then the resource of Northeast China was mentioned in Section 4 Materials and Methods (4.1. Plant Materials). Here we introduced that the tested materials were collected from four ecoregions in Northeast China, which are a collected sample of the Northeast China Soybean Germplasm Population (NECSGP, a representative sample of NECSGP). Here we do not call it as a breeding population (breeding materials in breeder’s hand), but a germplasm population or a sample of the NECSGP.
Comment 13: From this point to the end of the manuscript the reviewer addressed only issues of content, not language or grammar.
Line 54, “…accounting for 39.59% of the total cultivated land…” the authors should explain how this figure was imputed.
Answer: In the revised manuscript, we have added the calculation of 35.59% that “the proportion of saline-alkali land area accounted for cultivated land area in Northeast China is 1.141×107 hm2 / 2.882×107 hm2=35.95%.”
Comment 14: Line 91, “This procedure has…” the reviewer suggests the start of a new paragraph here. The foregoing discussion of methodology is interesting but should be abridged or moved to the Discussion section.
Answer: According to the comment, starting from the next sentence was changed to a new paragraph on the application of the identified QTLs in gene annotation. Here we put a brief introduction to RTM-GWAS procedure is mainly to let readers understand why we use this procedure in the present manuscript. While in Discussion section 3.2, there is a further discussion on the utilization and innovation of GASM-RTM-GWAS based on the results obtained from the present study.
Comment 15: Overall the Introduction contains too much technical information that should be abridged or moved to other sections such as Materials and Methods, Results, or Discussion.
Answer: Thanks for the comment. In the present study, the major success is the identification of 132 candidate genes with their 359 alleles, from which we understand ATI is a complex trait conferred by numerous genes with their multiple alleles in the germplasm population. Based on it, further results, such as evolutionary motivators were derived. This is because of the high power and efficiency of the GASM-RTM-GWAS procedure which is especially suitable for germplasm populations. The GASM-RTM-GWAS was developed based on RTM-GWAS procedure recently. We have met some often asked questions when submitting analogous manuscripts, so we prefer to put some recommendations of the procedures as introductions for Method, Results and Discussion sections.
Results
Comment 16: Results: Line 128, “…extremely significant…” the universally accepted term is “…highly significant…”.
Answer: Thanks for the suggestion. The suggested expression has been adopted.
Comment 17: Line 132, “…0.46~0.86, 0.46~0.86, 0.37~0.83, and 0.39~0.85…” to what do these ranges correspond to? The text indicates that these ranges were measured from different germplasm “sub-region” sources but these are not identified so are meaningless.
Answer: “…0.46~0.86, 0.46~0.86, 0.37~0.83, and 0.39~0.85” are the ATI ranges of soybean germplasm resources in different ecological sub-regions (ERA~ERD). The designation of different ecological sub-regions is in the Results section 2.1 (line 125, please see the note of Table 1, supplementary Figure 2 for details). At the end of the paragraph, “that means significant variation within each sub-region but not among them” was concluded.
Comment 18: Line 179, “…both effects…” reviewer presumes this refers to G and G×E effects? This should be stated in the text.
Answer: Yes, the term "both effects" in the manuscript means both G and G×E effects. It has been clarified in the manuscript.
Comment 19: Line 179-185, “For ATI, 98.43% (heritability value) of the phenotypic variation (PV) was explained by main effect genetic variation, in which 59.85% for the total R2 of 28 LC-major genes, 31.08% for 98 SC-major genes, in a total of 126 detected main effect genes explaining 90.94% PV, and the remained 98.43-90.94=7.49% PV might be explained by the collective unmapped minor genes. While the G×E genes explained only 2.80 % PV, which was very small.” The above results make us understand that alkali tolerance is a genetically complex trait.” The reviewer is not a quantitative geneticist so was not able to verify the validity of these estimates. If the authors have not already done so, these estimates should be confirmed by such an expert.
Answer: Thank you for your pertinent comment. This summarization of the genetic constitution analysis for a germplasm population was obtained from RTM-GWAS or GASM-RTM-GWAS procedure. Or in other words, this procedure characterizes with a relatively thorough identification of the genetic components in a population. We have published its application to different traits and populations, all these publications may confirm the reasonability of the procedure. These are as the following:
- Su Y, Zhang Z, He J, Zeng W, Cai Z, Lai Z, Pan Y, Hao X, Xing G, Wang W, et al: Gene–allele system of shade tolerance in southern China soybean germplasm revealed by genome-wide association study using gene–allele sequence as markers. Theor Appl Genet 2023, 136:152. https://doi.org/10.1007/s00122-023-04390-2
- Liu X, Li C, Cao J, et al. Growth period QTL‐allele constitution of global soybeans and its differential evolution changes in geographic adaptation versus maturity group extension. The Plant J, 2021, 1624:1643. https://doi.org/10.1111/tpj.15531
- Ali M J, Xing G, He J, Zhao T, Gai J, Detecting the QTL-allele system controlling seed-flooding tolerance in a nested association mapping population of soybean. The Crop J, 2020, 8(5): 781-792. https://doi.org/10.1016/j.cj.2020.06.008
- Fu M, Wang Y, Ren H, et al. Exploring the QTL–allele constitution of main stem node number and its differentiation among maturity groups in a Northeast China soybean population. Crop Sci, 2020, 1223: 1238. https://doi.org/10.1002/csc2.20024
- Fu M, Wang Y, Ren H, Du W, Wang D, Bao R, Yang X, Tian Z, Fu L, Cheng Y: Genetic dynamics of earlier maturity group emergence in south-to-north extension of Northeast China soybeans. Theor Appl Genet 2020, 133:1839-1857. https://doi.org/10.1007/s00122-020-03558-4
Comment 20: Lines 194-195, “The same is for below.” Table and figure legends should be complete and sufficient for each item described. Legends should not reference other legends.
Answer: Thanks for the comment. We have made detailed modifications to the notes of figures and tables to make each figure and table independent and complete.
Comment 21: Table 3, the presentation of data is confusing in this table. For the “main effects” and “allele effects” the number of alleles is depicted in different ways. The authors should reformat the table to be clearer.
Answer: Thanks for the comment. We have improved Table 3, so that the information of ATI gene-allele is clearer. This table was summarized from multiple tables with large sizes, to save space we put all the information on identified genes and alleles in a same table. For a concise summarization of the results on identified genes with their alleles, other journals accepted this short form, such as:
- Su Y, Zhang Z, He J, Zeng W, Cai Z, Lai Z, Pan Y, Hao X, Xing G, Wang W, et al: Gene–allele system of shade tolerance in southern China soybean germplasm revealed by genome-wide association study using gene–allele sequence as markers. Theor Appl Genet 2023, 136:152. https://doi.org/10.1007/s00122-023-04390-2
- Ali M J, Xing G, He J, Zhao T, Gai J, Detecting the QTL-allele system controlling seed-flooding tolerance in a nested association mapping population of soybean. The Crop J, 2020, 8(5): 781-792.
Comment 22: Line 210, “…breakthrough…” a more commonly used term by plant breeders would be “…transgressive…”.
Answer: Thanks for the suggestion. We have changed "breakthrough" to "transgressive" in accordance with your comments
Comment 23: Lines 226-227, “…according to the dissemination paths of soybean.” The references concerning the physical movement of Glycine germplasm in northeastern China from the Introduction should be reiterated here to document this statement and serve as the foundation for the following paragraphs.
Answer: Thanks for the comment. Yes, it has been reiterated in this sentence to serve as the foundation for the following paragraphs. This sentence is: “The NECSGP Gene-allele matrix was separated into its components of ERA−ERD. In Northeast China, ERD is the original eco-region at the south, while ERB is the derived eco-region based on ERD, and in turn, ERC and ERA were derived from ERB according to the dissemination paths of soybean.”
Comment 24: Lines 250-251, “…but there was ATI genetic structure changes…” The reviewer is confused about what the structural changes were; the authors should elaborate.
Answer: Thanks for the comment. “…but there was ATI genetic structure changes…” has been changed into “…but there was ATI gene-allele component changes…”. Here we use gene-allele component changes to replace genetic structure changes.
Comment 25: Lines 260-262, “The previous progress in alkali-tolerance was mainly a by-path without a scientific breeding plan.” Delete or move to the Discussion section.
Answer: According to the comment, this sentence has been deleted.
Comment 26: Line 272, Legend for Figure 2; “…number in different…” Reviewer believes the authors intend to say “…number of alleles specific to sub-region…”?
Answer: Thanks for the comment. In Figure 2, it mainly means “the allele number shared by eco-regions”. We have changed it.
Comment 27: Table 4, This table is full of figures, parentheses, varying units, etc. and is too confusing for the reviewer to follow. The authors should try to simplify the table to portray the basic message and move most of the details to an addendum.
Answer: Thanks for the comment. We have deleted the allele percentages in Table 4, and placed their calculation process in the main text to make the table simpler and clearer.
Comment 28: Line 328, “…(inadequate information in alkali-tolerance)…” Delete or explain.
Answer: According to the comment, it has been deleted.
Comment 29: Line 349, “…the key ones were nominated…” nominated how? By whom? Reviewer is confused by “nominated”.
Answer: According to the comment, this expression has been changed to “The key ATI genes were selected from the 132 identified ATI genes for future detailed studies according to the following criteria: …..”.
Comment 30: Lines 372-373, “…candidate genes with R2≥1.00% selected from 132 alkali tolerance candidate genes…” R2 of 1% seems quite small and arbitrary. Is there a precedent in the literature concerning the threshold for minimum correlation coefficient?
Answer: Thanks for the comment. In RTM-GWAS procedure, the identified genes/QTLs have their contribution R2 varied in a curve (such as in Figure 2C), a gene/QTL with its R2>inflection point is classified as a large contribution gene/QTL, otherwise as a small contribution one (such as in Table 3). Here 1% is roughly around the inflection point of the curve, in fact, 1% is not small among the 132 genes. Please refer to any of our published papers, such as:
He J, Meng S, Gai J, et al: An innovative procedure of genome-wide association analysis fits studies on germplasm population and plant breeding. Theor Appl Genet 2017, 130:2327. https://doi.org/10.1007/s00122-017-2962-9.
Discussion
Comment 31: Discussion: Line 437, “In the present study,…” New paragraph
Answer: According to the comment, it has been transferred to a new paragraph.
Comment 32: Line 451, “As indicated in the present study,” New paragraph
Answer: According to the comment, it has been transferred to a new paragraph.
Comment 33: Comments on the Quality of English Language
The English in this manuscript must be dramatically improved for this manuscript to be publishable.
Answer: We have invited an English-speaking colleague to help us to improve the writings of the manuscript.
